# Tryptophanyl-tRNA Synthetase 1 Signals Activate TREM-1 via TLR2 and TLR4

**DOI:** 10.3390/biom10091283

**Published:** 2020-09-06

**Authors:** Tram T. T. Nguyen, Hee Kyeong Yoon, Yoon Tae Kim, Yun Hui Choi, Won-Kyu Lee, Mirim Jin

**Affiliations:** 1Department of Health Sciences and Technology, GAIHST, Gachon University, Incheon 21999, Korea; dsthuytram@gmail.com (T.T.T.N.); isz03057@naver.com (H.K.Y.); kyt9204@naver.com (Y.T.K.); 2Department of Microbiology, College of Medicine, Gachon University, Incheon 21999, Korea; yunhuichoi@gmail.com; 3Lee Gil Ya Cancer and Diabetes Institute, Gachon University, Incheon 21999, Korea; 4New Drug Development Center, Osong Medical Innovation Foundation, Cheongju 28160, Korea; gre7@kbiohealth.kr

**Keywords:** tryptophanyl-tRNA synthetase 1, TREM-1, MyD88, TRIF, innate immunity

## Abstract

Tryptophanyl-tRNA synthetase 1 (WARS1) is an endogenous ligand of mammalian Toll-like receptors (TLR) 2 and TLR4. Microarray data, using mRNA from WARS1-treated human peripheral blood mononuclear cells (PBMCs), had indicated WARS1 to mainly activate innate inflammatory responses. However, exact molecular mechanism remains to be understood. The triggering receptor expressed on myeloid cells (TREM)-1 is an amplifier of pro-inflammatory processes. We found WARS1 to significantly activate TREM-1 at both mRNA and protein levels, along with its cell surface expression and secretion in macrophages. WARS1 stimulated TREM-1 production via TLR2 and TLR4, mediated by both MyD88 and TRIF, since targeted deletion of TLR4, TLR2, MyD88, and TRIF mostly abrogated TREM-1 activation. Furthermore, WARS1 promoted TREM-1 downstream phosphorylation of DAP12, Syk, and AKT. Knockdown of TREM-1 and inhibition of Syk kinase significantly suppressed the activation of inflammatory signaling loop from MyD88 and TRIF, leading to p38 MAPK, ERK, and NF-κB inactivation. Finally, MyD88, TRIF, and TREM-1 signaling pathways were shown to be cooperatively involved in WARS1-triggered massive production of IL-6, TNF-α, IFN-β, MIP-1α, MCP-1, and CXCL2, where activation of Syk kinase was crucial. Taken together, our data provided a new insight into WARS1′s strategy to amplify innate inflammatory responses via TREM-1.

## 1. Introduction

Innate inflammatory response is the first-line defense mechanism that occurs early in response to infection. Tryptophanyl-tRNA synthetase 1 (WARS1) is an essential housekeeping enzyme catalyzing the ligation of tryptophan to its cognate tRNA for protein synthesis [1,2]. Beyond its classical role, a non-canonical function of mammalian WARS1 is innate immune activation upon infection [3]. As soon as monocytes recognize invading pathogens, WARS1 is abundantly released from the cells, the secreted WARS1 functioning as an endogenous ligand of toll-like receptors (TLR) 2 and TLR4 [3,4]. The N-terminal domain of WARS1, which has been newly acquired during evolutionary process, is critical for the activation of TLRs. Truncated WARS1, lacking the 47 N-terminal amino acids, could neither activate TLR2/4 nor induce inflammatory responses. The WHEP (helix-turn-helix) motif, existing in the N-terminus, seems to be critical for TLR4 dimerization, to transmit signals in monocytes and macrophages. As a result, WARS1 rapidly induces massive cytokine and chemokine production, such as TNF-α, IL-6, IL-8, MIP-1α, IFN-α, and IFN-β [3,5].

Triggering receptors expressed on myeloid cells (TREM) form a family of cell surface receptors that are mainly expressed on innate immune cells, such as neutrophils, monocytes, and macrophages [6]. While TREM-1 represents the most active stimulator, TREM-2 serves as an inhibitor of inflammation [7]. Upregulation of TREM-1 expression had initially been demonstrated in inflammatory lesion caused by bacteria and fungal infection [8]. Although natural ligands for TREM-1 have not been clearly identified yet, TREM-1 is known to trigger and amplify inflammatory responses following TLR4 and TLR2 engagement [7]. Signaling through TLRs can be mainly categorized into two pathways: the common myeloid differentiation primary response protein 88 (MyD88)-dependent pathway and the MyD88-independent adapter-inducing interferon-β (TRIF) pathway [9]. TREM-1 activation in response to different TLR ligands occurs differently. In BM-derived macrophages, lipopolysaccharide (LPS) induces TREM-1 by activating TLR4-TRIF signaling. On the other hand, the effect of lipoteichoic acid is TLR2-MyD88-dependent [10]. Surface expression of TREM-1 is critical in signal transduction, for the amplification of inflammatory responses [11]. The intra-cytoplasmic tail of TREM-1 associates with the DNAX activating protein of 12 kDa (DAP12). DAP12 becomes phosphorylated at its immune receptor tyrosine-based activation motif (ITAM) region, providing a docking site for protein tyrosine kinase 70 (ZAP70) and spleen tyrosine kinase (Syk) [10]. The tyrosine kinase Syk/ZAP70 activates phosphoinositide 3-kinase (PI3K) and extracellular signal-regulated kinases (ERK) pathways, which then lead to a positive feedback that activates signaling and transcription factors, including p38 mitogen-activated kinases (p38 MAPK)/ERK pathway, and NF-κB (p50–p65), which are responsible for the production of various pro-inflammatory cytokines and chemokines [11,12].

In this study, we investigated how WARS1 facilitates TLR and TREM-1 signaling to exert innate inflammatory responses in macrophages. WARS1 induced TREM-1, thus amplifying TREM-1 signaling, and added a positive feedback loop in TLR cascade via MyD88 and TRIF.

## 2. Materials and Methods

### 2.1. Microarray Screening

Total RNA from cultured human peripheral blood mononuclear cell (PBMCs), grown under normal condition or treated with 100 nM WARS1, was isolated using the RNeasy^®^ mini kit (Qiagen GmbH, Hilden, Germany) according to the manufacturer’s protocol. Quantity and quality of RNA were assessed by NanoDrop™ 1000 spectrophotometer (Thermo Fisher Scientific, Inc., Waltham, MA, USA) and BioAnalyzer™ 2100 (Agilent Technologies, Santa Clara, CA, USA). Microarray analysis was performed using GeneChip™ Human Gene 2.0 ST Array (Affymetrix Inc., Santa Clara, CA, USA). Briefly, RNA samples were amplified using 500 ng of total RNA, and after biotin labeling, the labeled RNA was hybridized to the Affymetrix Human Gene 2.0 ST array using the Affymetrix hybridization system; array wash and staining were performed using Fluidics Station 450. Following this, Affymetrix scanner 3000 7G was used to extract signal data. Analysis of DNA chip image was normalized using the Robust Multi-array Average (RMA) procedure; the signal intensity value of the spot obtained through scanning was displayed as a scatter plot. Distribution of all genes, obtained from the experiment, was displayed as a histogram, which was then compared with the distribution of normalized data to confirm accuracy of the experiment. Finally, genes showing ±1.5–fold change and *p* ≤ 0.05 were selected using GeneSpring GX 13.1.1, and the differentially expressed genes (DEG) were analyzed. A hierarchical clustering heatmap was generated using Cluster 3.0 and TreeView software.

### 2.2. Antibodies and Chemicals

Immunoblot analysis was performed as described earlier [13], using the following antibodies: rabbit anti-TREM-1 (Abcam, USA), rabbit anti-MyD88 (Cell Signaling, USA), rabbit anti-TRIF (Thermo Fisher Scientific, USA), mouse anti-Syk (Abcam, USA), rabbit anti-p-Syk (Thermo Fisher Scientific, USA), rabbit anti-DAP12 (Cell Signaling, USA), rabbit anti-AKT (Cell Signaling, USA), rabbit anti-p-AKT (Cell Signaling, USA), mouse anti-TLR4 (Santa Cruz, USA), rabbit anti-TLR2 (EMD Millipore, USA), rabbit anti-p38 (Cell Signaling, USA), rabbit anti-p-p38 (Cell Signaling, USA), rabbit anti-ERK (Cell Signaling, USA), rabbit anti-p-ERK (Cell Signaling, USA), rabbit anti-IκB (Cell Signaling, USA), rabbit-anti-p-IκB (Cell Signaling, USA), mouse anti-C/EBP (Santa Cruz, USA), mouse anti-β-actin (Santa Cruz, USA), and rabbit anti-4G10 (Sigma-Aldrich, USA). Either horseradish peroxidase-conjugated goat anti-rabbit antibody or goat anti-mouse antibody (Cell Signaling, USA) was used as a secondary antibody. R406 was purchased from Selleckchem.

### 2.3. Cell Culture

All cell lines were grown in Dulbecco’s modified Eagle’s medium supplemented with 10% fetal bovine serum and 1% penicillin-streptomycin, in 5% CO_2_ at 37 °C. THP-1 cells were differentiated for 3 days with phorbol-12-myristate-13-acetate (50 ng/mL), before use. 

### 2.4. Recombinant Protein Purification

A series of human WARS1, murine WARS1 (FL-WARS1), and N-terminal 85 aa-deleted murine WARS1 (Δ85-WARS1) were cloned in pET-28a (His-tag) vector. WASR1 mutants were generated by site-directed mutagenesis (Agilent Technologies, USA). The recombinant proteins were overexpressed in *Escherichia coli* Rosetta2 (DE3) strain. The bacterial cells were lysed by three rounds of sonication (10 s duration at 50 s intervals). His-tagged proteins in the supernatants were purified using a HisTrap HP column (GE Healthcare, USA), followed by endotoxin removal using Amicon^®^ Ultra-4 Centrifugal Filter Unit (Millipore, Ireland). All used proteins were confirmed by Coomassie staining and tested for endotoxin levels using Pierce™ LAL Chromogenic Endotoxin Quantitation Kit (Thermo Scientific, USA) (Appendix A). WARS1 was treated at 50–200 nM for 8 h into J774.1 cells.

### 2.5. RNA Interference

siRNAs targeting TLR4, TLR2, MyD88, TRIF, and TREM-1, and the universal negative control siRNA were purchased from Santa Cruz. The siRNAs were transfected using a Lipofectamine RNAiMax reagent (Invitrogen, USA), according to the manufacturer’s instructions.

### 2.6. GST Pulldown Assay

The glutathione *S*-transferase (GST)–WARS1 fusion protein was expressed in *Escherichia coli* Rosetta2 (DE3) strain and purified with glutathione-Sepharose 4B beads (GE Healthcare, USA) according to the manufacturer’s instructions. For the in vitro binding assay, cell lysates were incubated with either GST or GST-WARS1 for 2 h at 4 °C in cell lysis buffer. The samples were washed four times with lysis buffer, and the bound protein was detected by immunoblot assay. 

### 2.7. Quantification of RNA

Quantitative real-time PCR (qRT-PCR) experiments were conducted using an iQ5 multicolor real-time PCR detection system (Bio-Rad Laboratories, Hercules, CA, USA), as reported previously [14]. The gene-specific primer sequences are presented in Appendix A.

### 2.8. Flow Cytometry

For surface staining, cells were stained with an anti-rabbit-TREM-1 antibody (Abcam) for 30 min. Cells were washed with phosphate-buffered saline (PBS), and stained with an anti-rabbit fluorescein isothiocyanate (FITC) (Cell Signaling) at room temperature for 30 min. Cell were then washed again with PBS, fixed with 1.5% formaldehyde for 20 min under room temperature, and resuspended in FACS buffer (PBS with 5% fetal calf serum), before analysis, using a flow cytometer (BDFACS Calibur, USA), as described previously. Forward and side scatter gating with 10^5^ cells were applied [14].

### 2.9. Immunofluorescence Assay

J774.1 cells, grown on cover slides, were fixed in 4% paraformaldehyde in PBS for 10 min and then permeabilized with 0.1% Triton X-100 in PBS for 10 min at room temperature. After three washes with PBS, the fixed cells were blocked with 1% bovine serum albumin (BSA) in PBS for 1 h at room temperature. The cells were then incubated with an indicated antibody overnight at 4 °C. After three washes with PBS, the cells were incubated with tetramethylrhodamineisothiocyanate (TRITC)-conjugated donkey anti-rabbit IgG for 1 h at room temperature. They were then counterstained with 4′,6-diamidino-2-phenylindole (DAPI) to label the nuclei. After three washes with PBS, cells were analyzed under the Zeiss LSM 700 laser confocal microscope (Carl Zeiss, Inc., Thornwood, NY, USA).

### 2.10. ELISA

The levels of IL-6, TNF-α, IFN-β, MIP-1α, MCP-1, and CXCL2 in cell culture supernatants were determined using commercially available ELISA kits, according to the manufacturer’s protocol. ELISA kits for mouse TNF-α (R&D systems, USA, DY410), mouse MIP-1α (R&D systems, USA, DY450), mouse IL-6 (BioLegend, USA, 431304), mouse MCP-1 (BioLegend, USA, 432704), mouse CXCL2 (R&D systems, USA, DY452), and mouse IFN-β (R&D systems, USA, DY8234) were used appropriately.

### 2.11. WST Assay

Cells were seeded into 12-well plates. Cell viability was determined using 30 µL of water-soluble tetrazolium salt (WST) in each well, with the plate incubated at 37 °C for 1 h. After shaking the plate for 1 min, the aqueous layer in each well was transferred to a 96-well plate, and absorbance at 450 nm was measured.

### 2.12. Statistical Analysis

Data are presented as mean ± standard deviation (SD). Student’s *t* test was used for statistical analysis. The asterisks in the figures indicate significant differences (* *p* < 0.05; ** *p* < 0.01; *** *p* < 0.001; ns, not significant). Experiments were repeated three times.

## 3. Results

### 3.1. WARS1 Mainly Activated Innate Inflammatory Response-Related Gene Expression in Human PBMCs

To gain insight into the function of WARS1 in the inflammatory cascade, we performed a microarray analysis using mRNA from WARS1-treated human PBMCs, as described in Materials and Methods above. Statistically significant candidates were determined by ±1.5–fold change in expression (Figure 1a). Heatmap analysis of the scanned array data revealed 30 genes significantly upregulated at transcriptional levels while only 3 genes were significantly downregulated by WARS1 (Figure 1b). The genes that were differentially mediated in WARS1-treated PBMCs encoded molecules with well-known functions in inflammation of phagocytes, namely cytokines (IL-1β, IL-8, and OSM), chemokines (CCL2, CCL22, CCL24, and CXCL5) and receptors (TREM-1 and FCRGT) (Figure 1c). Ontological data revealed most of the genes regulated by WARS1 to be involved in immune response, inflammation, and cell proliferation (Figure 1d). The differential of several genes in this microarray was confirmed by qRT-PCR using a murine macrophage cell line J774.1. Similar to LPS, WARS1 (50–200 nM) induced IL-1β, TIMP-1, and CXCL2 (IL-8 homolog) mRNA expression in a dose-dependent manner (Appendix A).

### 3.2. WARS1 Activated TREM-1 Expression

To further investigate the molecular mechanisms underlying inflammatory responses by WARS1, we selected TREM-1, one of the pro-inflammatory receptors that can directly amplify TLR signaling in macrophages, from microarray data [6]. The mRNA levels of TREM-1, not TREM-2, were significantly increased in WARS1-treated J774.1 cells in time-dependent (Figure 2a) and dose-dependent manner (Figure 2b). LPS served as a positive control [15]. Protein levels of intracellular TREM-1, as well as of the secreted soluble form, were remarkably elevated by WARS1 after 8 h treatment (Figure 2c). Consistently, mRNA and protein levels of TREM-1 in human THP-1 and murine RAW267.4 macrophages showed similar response patterns (Appendix A). The WARS1-mediated effect was unaffected by concurrent addition of LPS inhibitor Polymyxin B, although the latter efficiently blocked LPS response (Figure 2d). It was noteworthy that heat-inactivated WARS1 could not induce TREM-1 mRNA expression. LPS was still active at temperatures used to denature WARS1 (Figure 2d). Furthermore, LAL test showed very low levels of endotoxin to exist in all the used proteins (<0.03 EU). These data clearly ruled out any potential LPS contamination, demonstrating WARS1 to indeed represent a TREM-1 activator. As shown in Figure 2e, WARS1 stimulated membrane-bound TREM-1 expression comparable to that by LPS treatment. Using immunofluorescence analysis, we confirmed TREM-1 membrane receptor to be induced by WARS1 (Figure 2f). Taken together, these results indicated WARS1 to induce TREM-1 surface expression, as a result of TREM-1-transcription activation.

### 3.3. WARS1 Activated TREM-1 Expression via TLR4 and TLR2

We next examined whether WARS1 could directly bind to and mediate TREM-1 activation, using an in vitro glutathione S- transferase (GST) pulldown assay. No direct interaction between WARS1 and TREM-1 was observed (Figure 3a). To check whether TLR2 and/or TLR4 were responsible for the amplification of TREM-1, we created an N-terminal ∆85 aa (lacking WHEP domain) murine-WARS1 mutant (Figure 3b, Appendix A), which was inactive in TLR signaling. Figure 3c suggests that cells treated with 100 nM WARS1 showed a strong induction of TREM-1 transcriptional activity, although not so for TREM-2. Compared to wild-type controls, ∆85 mutant showed no effect on TREM-1 mRNA and protein levels (Figure 3c,d). Knockdown of TLR2 or TLR4, or both TLR2 and TLR4, almost completely diminished WARS1-induced TREM-1 mRNA and protein expression (Figure 3e,f). These data suggested that WARS1 activated TREM-1 through TLR2 and TLR4 in macrophages.

### 3.4. WARS1 Facilitated Both TRIF and MyD88 for TREM-1 Expression

We further investigated MyD88 and TRIF signaling. First, WARS1 elevated MyD88 transcription in a time and dose-dependent manner (Figure 4a,b). WARS1 induced TRIF mRNA levels after as early as 1 h and the activation was dose-dependent (Figure 4c,d). Protein levels of MyD88 and TRIF were significantly upregulated by WARS1 treatment (Figure 4e,f). Finally, considering that WARS1 monitored the activation of well-known molecular effectors in TLRs signaling, comparable to LPS (phosphorylation of p38 MAPK, ERK, IκB, and the transcription factor C/EBP (Appendix A)), we wondered whether the signal transduction pathway induced by WARS1 was dependent on MyD88 and/or TRIF. We used siRNA to knockdown MyD88 or TRIF and then treated with WARS1 subsequently. MyD88 or TRIF-silenced cells showed reduced protein levels of TREM-1 upon WARS1 activation (Figure 4g,h). There was some overlap between MyD88 and TRIF signaling; our data indicated that WARS1 activated both molecules and their common downstream pathway, including p38 MAPK, ERK, and IκB phosphorylation (Figure 4g,h). Taken together, these data demonstrated WARS1 to positively regulate TREM-1 expression by inducing both MyD88 and TRIF pathways.

### 3.5. WARS1 Activated TREM-1 Signaling

To understand whether WARS1 has an effect on TREM-1 downstream signaling pathway, activation of DAP12, Syk, and AKT was examined. Syk and DAP12 mRNA levels were not altered after WARS1 treatment (Figure 5a,b). However, WARS1 was able to induce phosphorylation of DAP12 and Syk, as well as their downstream AKT (Figure 5c). The stimulatory effects of WARS1 on macrophages could be significantly blocked by a Syk-specific pharmacological inhibitor R406. As can be seen in Figure 5d, Syk and its downstream AKT phosphorylation was inhibited, indicating the effect of R406 on TREM-1 signaling. Interestingly, MyD88 and TRIF expression, as well as phosphorylation events of their downstream molecules, showed reduced activation by WARS1 in presence of R406. Treatment of the same concentration of R406 showed no toxicity (Appendix A). Similarly, lack of responses from MyD88 and TRIF, as well as p38 MAPK, IκB, and ERK phosphorylation to WARS1 in TREM-1-silenced macrophages was additionally confirmed (Figure 5e). Thus, TREM-1 signaling was critical for the amplification of MyD88 and TRIF pathways upon WARS1 treatment.

### 3.6. WARS1 Promoted Cytokine and Chemokine Production by the Cooperation of MyD88-TRIF-TREM-1 Signaling

To understand interactions among MyD88, TRIF, and TREM-1 signaling in inflammatory responses by WARS1, each individual signaling pathway was blocked using siRNAs, and R406 was used for Syk inhibition. The levels of IL-6, TNF-α, IFN-β, MIP-1α, MCP-1, and CXCL2 were determined in the supernatants of WARS1-stimulated macrophages (Figure 6a–d). We observed cooperation among the pathways for the effective release of IL-6, TNF-α, and MCP-1 (Figure 6a–d). The amount of WARS1-induced IFN-β, CXCL2, and MIP-1α secretion was significantly lower in cells lacking TRIF, though not in MyD88-silenced cells (Figure 6b). Amplification of TREM-1/Syk signaling was necessary for the release of all cytokines and chemokines stimulated by both MyD88 and TRIF (Figure 6c,d). Collectively, these data indicated MyD88 and TRIF pathways to separately or cooperatively activate particular set of genes, the synergy of which was mediated by TREM-1 signaling.

## 4. Discussion

We recently found human WARS1 to be an endogenous TLR ligand for monocyte/macrophage activation against infection [3]. In this study, we further characterized WARS1 as an innate immune activator by understanding the way it activates innate inflammatory responses via TREM-1 in macrophages. During evolutionary process, WARS1 seems to have been optimized, and designed to efficiently perform immunological function, including rapid, nonspecific, and immense responses during acute phase of infection. First, WARS1 acts rapidly; it is promptly secreted within several minutes, following infection, without de novo synthesis. Second, WARS1 is secreted regardless of the type of pathogens, including G(+) or G(−) bacteria, virus, and fungi, indicating non-specificity, covering all invading pathogens [3]. Third, WARS1 developed expansion mechanism for strong cytokine and chemokine production using diverse routes, acting on not only MyD88 but also TRIF, and additionally activating TREM-1 signaling. Upregulation of TREM-1 expression was initially demonstrated in neutrophils and monocytes in response to lipopolysaccharide and other microbial products [11]. Interestingly, crosslinking of only TREM-1 induced tolerable cellular activation and cytokine secretion. However, co-ligation of TREM-1 with pathogen-associated molecular pattern (PAMP)-mediated activation resulted in much more pro-inflammatory cytokine production than the sum of the responses induced by either TREM-1 or PAMPs alone [16,17,18]. We confirmed that WARS1-induced TREM-1 activation employs both TLR2/4 and MyD88/TRIF for its synergism. Conversely, inhibition of TREM-1 by siRNA or of TREM-1 signaling by R406, which binds to the ATP-binding pocket and inhibits Syk kinase activity, reduced both WARS1-induced MyD88 and TRIF activation, suggesting a synergistic interaction between TREM-1 and WARS1-mediated TLR2/TLR4 signaling. The cooperative signaling merged with p38, ERK, and NF-κB activation, completing the amplification of inflammatory loop (Figure 7). Finally, our data showed CXCL2, MIP-1α, and IFN-β secretion to depend on TRIF signaling. Both MyD88 and TRIF pathways are necessary for IL-6, TNF-α, and MCP-1 release (Figure 6). TREM-1/Syk is crucial for WARS1-induced secretion of all the cytokines and chemokines. Taken together, our study proposed WARS1 to have developed a unique amplification mechanism for activation of innate immune responses, especially massive secretion of diverse chemokines and cytokines, by facilitating MyD88 and TRIF via both TLR2 and TLR4, which are amplified by TREM-1, to efficiently deal with infection. Interestingly, WARS1 induced more secretion of the soluble form of TREM-1 (sTREM-1) as compared to LPS (Figure 2). In addition to the 27 kDa membrane receptor, TREM-1 exists as a 15~17.5 kDa soluble form in biological fluid during infection or inflammation. However, little is known about the origin and function of sTREM-1. LPS induces sTREM-1 secretion, but only at high doses [6,12]. We did not detect sTREM-1 after treating macrophages with 500 ng/mL LPS, whereas WARS1 treatment at 50 nM (~2.7 µg/mL) was sufficient to activate TREM-1 on the cell surface, as well as activating sTREM-1. Hence, WARS1 may show high sensitivity upon TREM-1 signaling. Further studies are necessary to reveal the functional relevance of WARS1-induced sTREM-1 secretion.

Sepsis is a life-threatening systemic inflammatory syndrome against severe infection. Dysregulated host immune response during the acute phase of sepsis is characterized by an exaggerated systemic release of cytokines and chemokines (cytokine storm), which in turn can cause tissue damages and multi-organ failure [19,20,21]. Consistent with our experimental findings, in clinical settings, we found WARS1 to be greatly secreted into the blood of patients with sepsis more than in those with sterile inflammatory disorder [3]. A recent retrospective study indicated that the high levels of WARS1 are correlated with disease severity in patients with sepsis, who have been admitted in intensive care units [22]; accumulated data have suggested WARS1 to be an upstream molecule inducing cytokine storm [3,23], indicating a therapeutic target for hyper-inflammatory septic death. TREM-1 was identified as a key player of hyper-inflammation in sepsis [18]. Inhibition of TREM-1 with chemically synthesized peptides, namely LR12, LR17, and LP17, limits the production of pro-inflammatory cytokines, such as IL-1β, TNF-α, and IL-8 [16,24,25,26]. LR12, a peptide consisting of 12 amino acids, exhibited protective effects and improved survival in pigs and mice [7,18]. LR12 was chosen as the first drug candidate to address the possibility of TREM-1 in clinical stage development [27]. These results greatly encouraged us to test the potential of WARS1 as a therapeutic target for sepsis. WARS1 works as an upstream regulator for TREM-1, as well as other cytokines and chemokines. WARS1 exhibits stronger efficacy, by working on both TLRs and TREM-1 signaling. Blocking TREM-1 could lead to compensatory effects in other routes, which can regulate the same genes; such things happen regularly in TLR signaling [28,29,30]. Therefore, upstream targeting of WARS1 creates fewer opportunities for escape and bypass of redundant TLR-downstream pathways. Collectively, secreted WARS1 may represent an expected sepsis drug target at the apex of the TLRs/TREM-1 signaling cascade.

## 5. Conclusions

This study demonstrated, for the first time, how WARS1 acquired potent pro-inflammatory properties as a ligand of TLR2/4. WARS1 induced downstream mediators of TLRs, namely MyD88 and TRIF, resulting in elevated secretion of IL-6, TNF-α, IFN-β, MIP-1α, MCP-1, and CXCL2. Moreover, WARS1 upregulated TREM-1, thus amplifying TREM-1/Syk signaling, and added a positive feedback loop in the TLR cascade through MyD88 and TRIF. Collectively, WARS1-mediated TREM-1 could be an attractive target for sepsis treatment owing to its ability to modulate innate immune responses. 

## Figures and Tables

**Figure 1 biomolecules-10-01283-f001:**
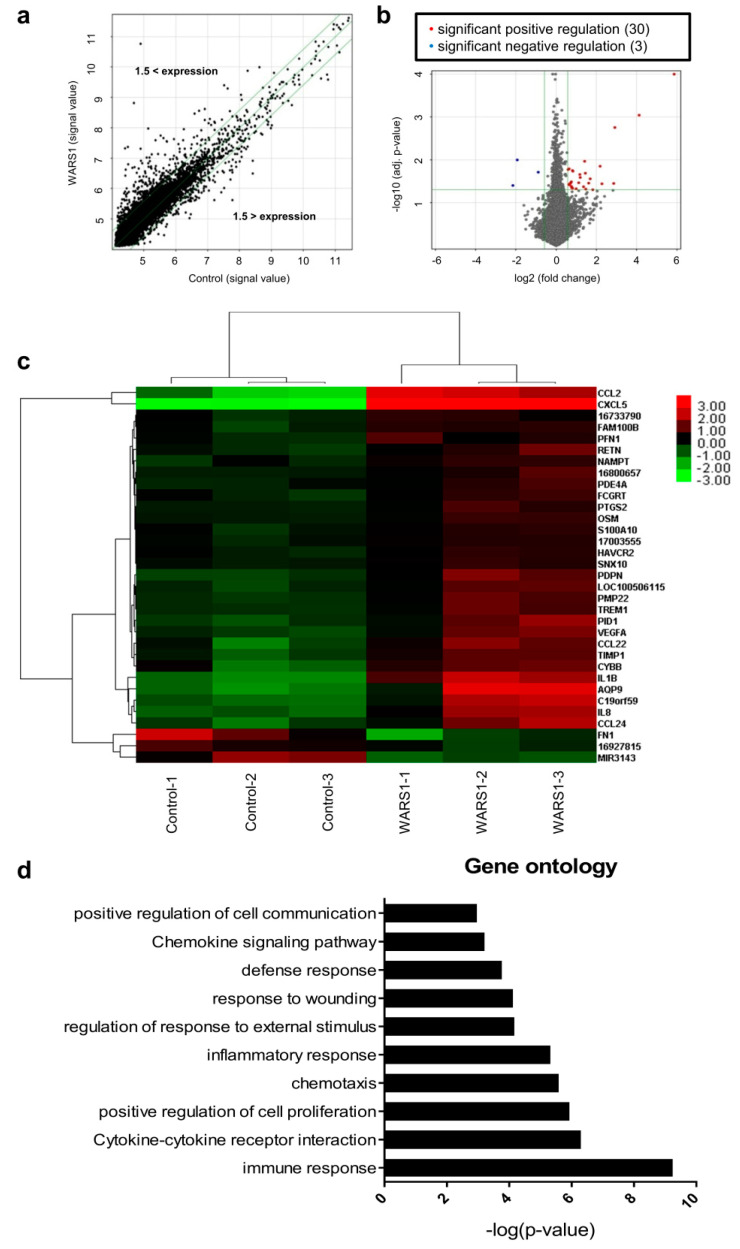
WARS1 activated innate inflammatory response genes. (**a**) Scatter plot of total gene expression of PBMCs treated with 100 nM hWARS1 for 24 h. Green line indicates fold change ± 1.5 on log_2_ scale. Each dot represents a gene. Genes with up-regulated expression in WARS1-treated cells are located above the green line (ratio ≥ 1.5–fold), whereas genes with down-regulated expression in WARS1-treated cells are located below the green line (ratio ≤ 1.5–fold). (**b**) Volcano plot of differentially expressed genes in WARS1-treated cells than in normal conditions. Red dots indicate significantly up-regulated genes (log_2_ fold change ≥ 1.5, *p* < 0.05) and blue dots indicate significantly down-regulated genes (log_2_ fold change ≤ 1.5, *p* < 0.05). (**c**) Heat map of 33 genes showing significant differences in expression. Red, upregulated genes; black, unchanged genes; green, downregulated genes. (**d**) Gene ontology (GO) analysis for microarray-based differentially expressed genes was performed using DAVID bioinformatics tool 6.7.

**Figure 2 biomolecules-10-01283-f002:**
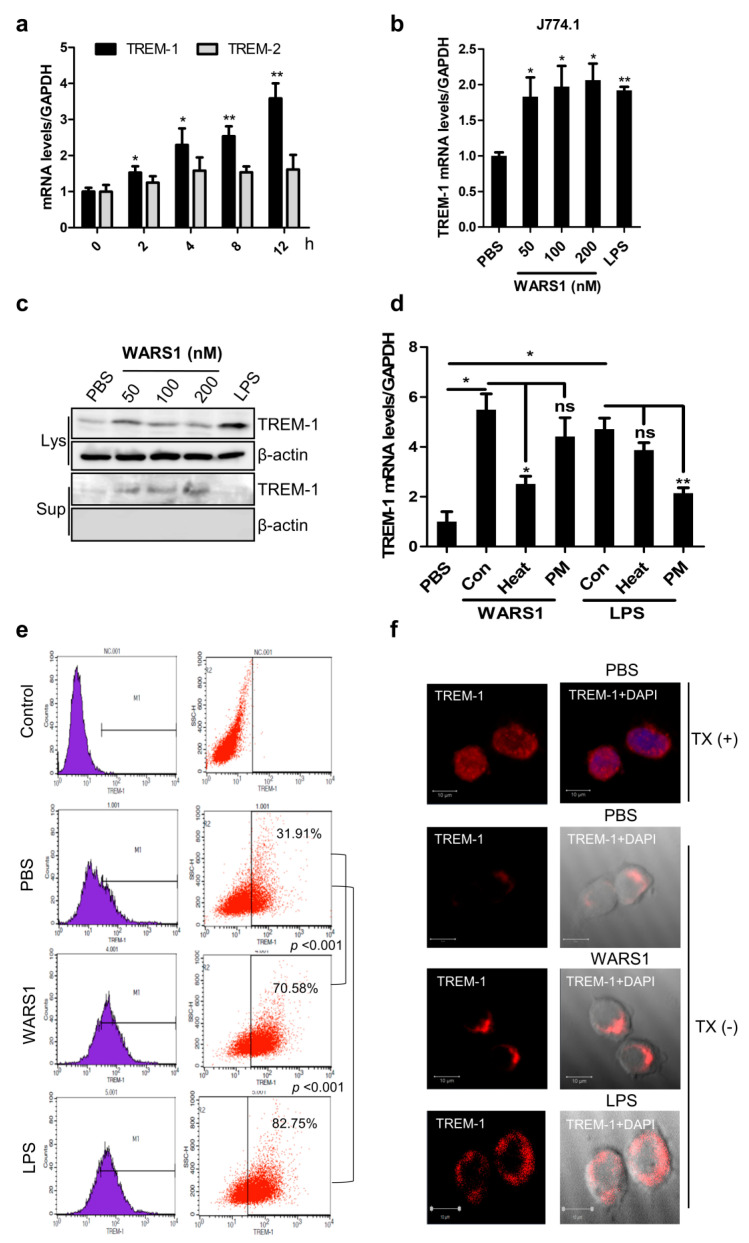
WARS1 activated TREM-1 expression. (**a**) J774.1 cells were treated with 100 nM WARS1 and then mRNA levels of TREM-1 and TREM-2 were analyzed by qRT-PCR at the indicated time points. (**b**) mRNA levels of TREM-1 in J774.1 cells treated with increasing doses of WARS1 or 500 ng/mL LPS for 8 h were determined by qRT-PCR. (**c**) Cell lysate (Lys) and cell supernatant (Sup) levels of TREM-1 in J774.1 cells, treated with increasing doses of WARS1 or 500 ng/mL LPS for 8 h, were determined by immunoblotting. (**d**) Polymyxin B (25 μg/mL) had no inhibitory effect on WARS1-induced TREM-1 expression in J774.1 cells, whereas the effects of LPS were efficiently blocked. In addition, samples were heated for 30 min at 80 °C before stimulation. No effect, due to heating, was observed for LPS, whereas inhibition was indeed observed for heated WARS1. TREM-1 levels were determined by qRT-PCR after 12 h of stimulation (Con, control; Heat, 30 min at 80 °C; PM, Polymycin B). (**e**) J774.1 cells were left untreated or treated with 100 nM WARS1 or 500 ng/mL LPS for 16 h; thereafter, TREM-1 expression on cell surface was detected by flow cytometry. Data represent averages from three independent experiments. The asterisks indicate significant difference (* *p* < 0.05; ** *p* < 0.01;). (**f**) J774.1 cells were seeded onto coverslips and then treated with PBS or 100 nM WARS1 or 500 ng/mL LPS for 16 h. Immunofluorescence staining was performed to detect TREM-1 (red) and nuclei (blue) using DAPI.

**Figure 3 biomolecules-10-01283-f003:**
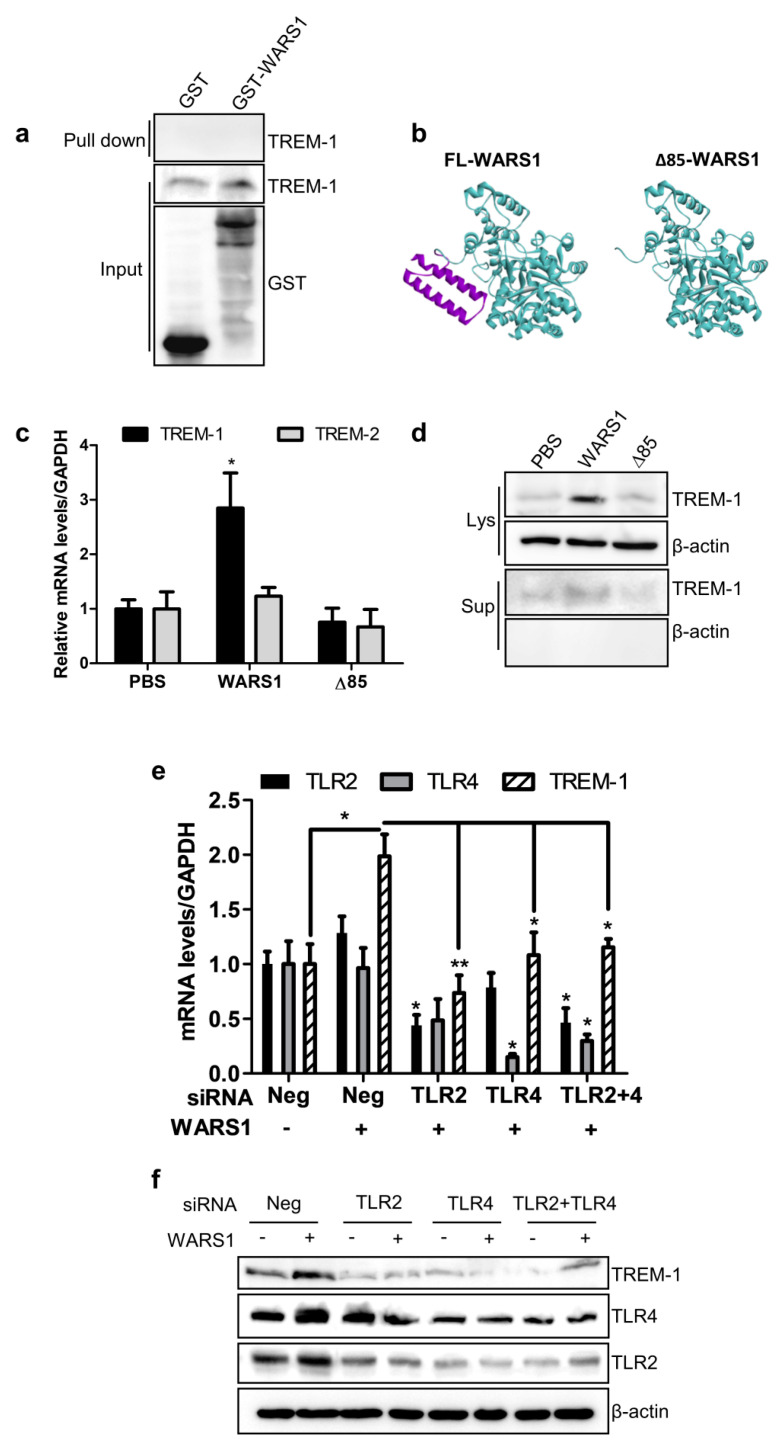
WARS1 up-regulated TREM-1 expression via TLR2 and TLR4. (**a**) J774.1 total cell lysates were incubated with either GST or GST-WARS1. After pulldown by GST beads, bound protein was detected by immunoblot analysis with an anti-TREM-1 antibody. (**b**) Crystal structure of the murine WARS1 monomer (left) and murine N-terminal ∆85 aa-deleted inactive mutant WARS1 (right). (**c**) J774.1 cells were treated with PBS, 100 nM WARS1, or 100 nM N-terminal ∆85 mutant WARS1, for 8 h and then mRNA levels of TREM-1 and TREM-2 were analyzed by qRT-PCR. (**d**) Cell lysate (Lys) and cell supernatant (Sup) levels of TREM-1 in J774.1 cells, treated as described in panel (**c**), were determined by immunoblotting. (**e**) J774.1 cells were transfected with 100 nM negative, or TLR2 or TLR4 siRNA, or a combination of 50 nM TLR4 and 50 nM TLR2 siRNAs; 72 h after transfection, cells were treated with PBS or 100 nM WARS1 for 8 h, and then mRNA levels were determined by qRT-PCR. (**f**) J774.1 cells, treated as described in panel (**e**), were immunoblotted with the indicated antibodies. (*, *p* < 0.05, **, *p* < 0.01).

**Figure 4 biomolecules-10-01283-f004:**
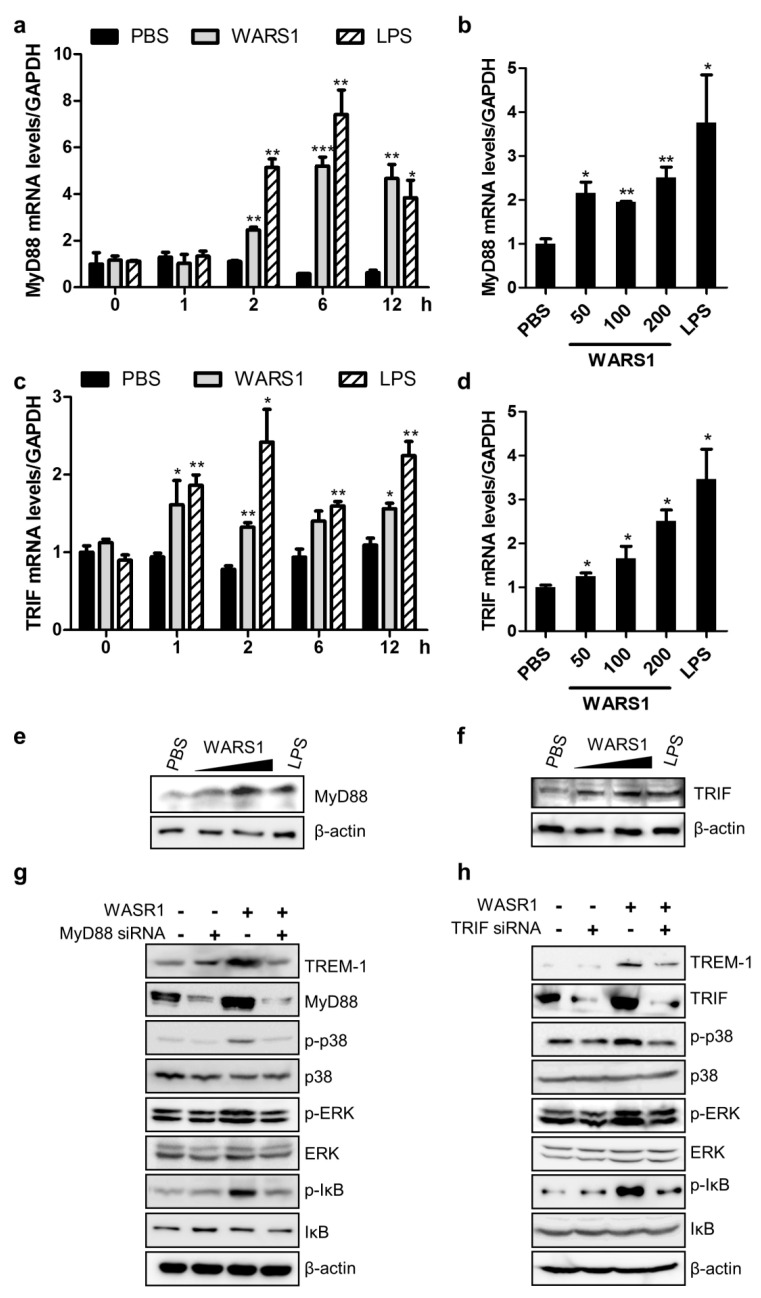
WARS1 activated both MyD88 and TRIF-dependent signaling pathway. (**a**) J774.1 cells were treated with PBS, 100 nM WARS1, or 500 ng/mL LPS, and then mRNA levels of MyD88 were analyzed by qRT-PCR at tfhe indicated time points. (**b**) mRNA levels of MyD88 in J774.1 cells treated with increasing doses of WARS1 or 500 ng/mL LPS for 8 h were determined by qRT-PCR. (**c**) J774.1 cells were treated with PBS, 100 nM WARS1, or 500 ng/mL LPS, and then mRNA levels of TRIF were analyzed by qRT-PCR at the indicated time points. (**d**) mRNA levels of TRIF in J774.1 cells treated with increasing doses of WARS1 or 500 ng/mL LPS for 8 h were determined by qRT-PCR. (**e**) J774.1 cells treated as described in panel **b** were immunoblotted with the indicated antibodies. (**f**) J774.1 cells treated as described in panel **d** were immunoblotted with the indicated antibodies. (**g**) J774.1 cells were transfected with 100 nM the negative or MyD88 siRNAs; 72 h after transfection, cells were treated with PBS or 100 nM WARS1 for 8 h, and protein expression levels were determined by immunoblotting. (**h**) J774.1 cells were transfected with 100 nM negative or TRIF siRNAs; 72 h after transfection, cells were treated with PBS or 100 nM WARS1 for 8 h, and protein expression levels were determined by immunoblotting. (*, *p* < 0.05, **, *p* < 0.01; *** *p* < 0.001).

**Figure 5 biomolecules-10-01283-f005:**
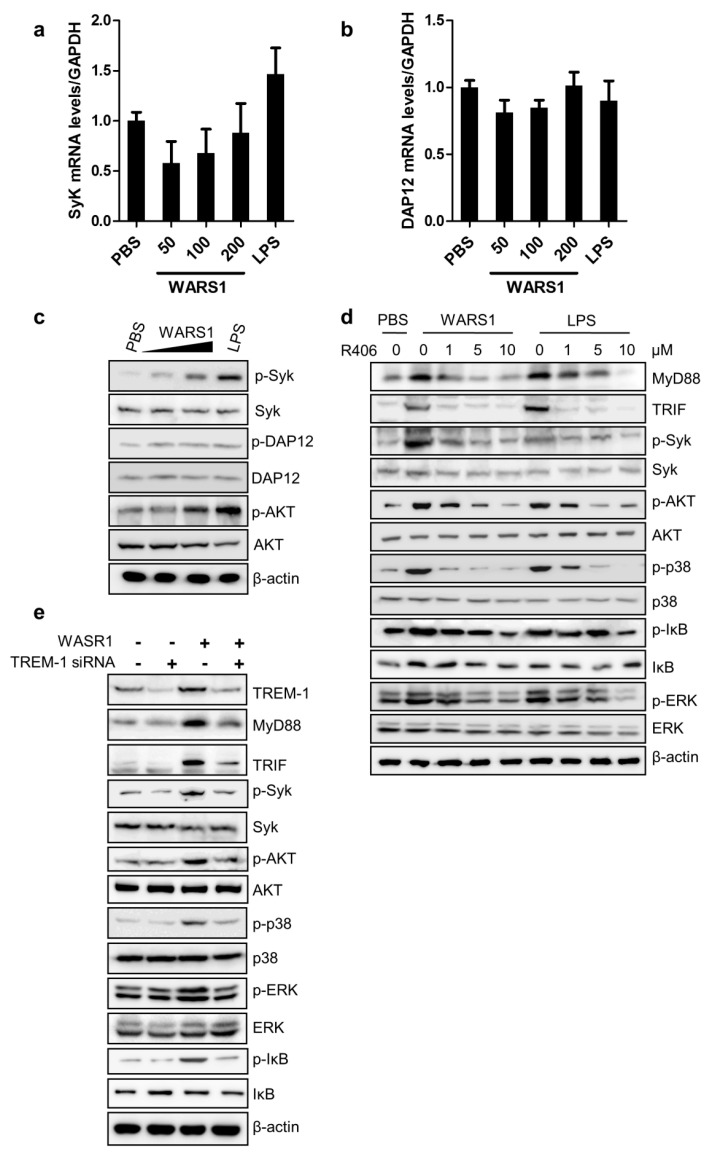
WARS1 induced TREM-1 pathway to amplify the TLR signaling. (**a**,**b**) mRNA levels of (**a**) Syk and (**b**) DAP12 in J774.1 cells treated with increasing doses of WARS1 or 500 ng/mL LPS for 8 h were determined by qRT-PCR, respectively. (**c**) J774.1 cells were treated with increasing doses of WARS1 or 500 ng/mL LPS for 8 h, and then protein levels were determined by immunoblotting using indicated antibodies. Phosphorylation levels of DAP12 were detected by a G410 antibody. (**d**) J774.1 cells were treated simultaneously with increasing doses of the Syk inhibitor (R406) and PBS, 100 nM WARS1, or 500 ng/mL LPS for 8 h, and then protein expression levels were determined by immunoblotting. (**e**) J774.1 cells were transfected with 100 nM negative or TREM-1 siRNAs; 72 h after transfection, the cells were treated with PBS or 100 nM WARS1 for 8 h, and protein expression levels were determined by immunoblotting.

**Figure 6 biomolecules-10-01283-f006:**
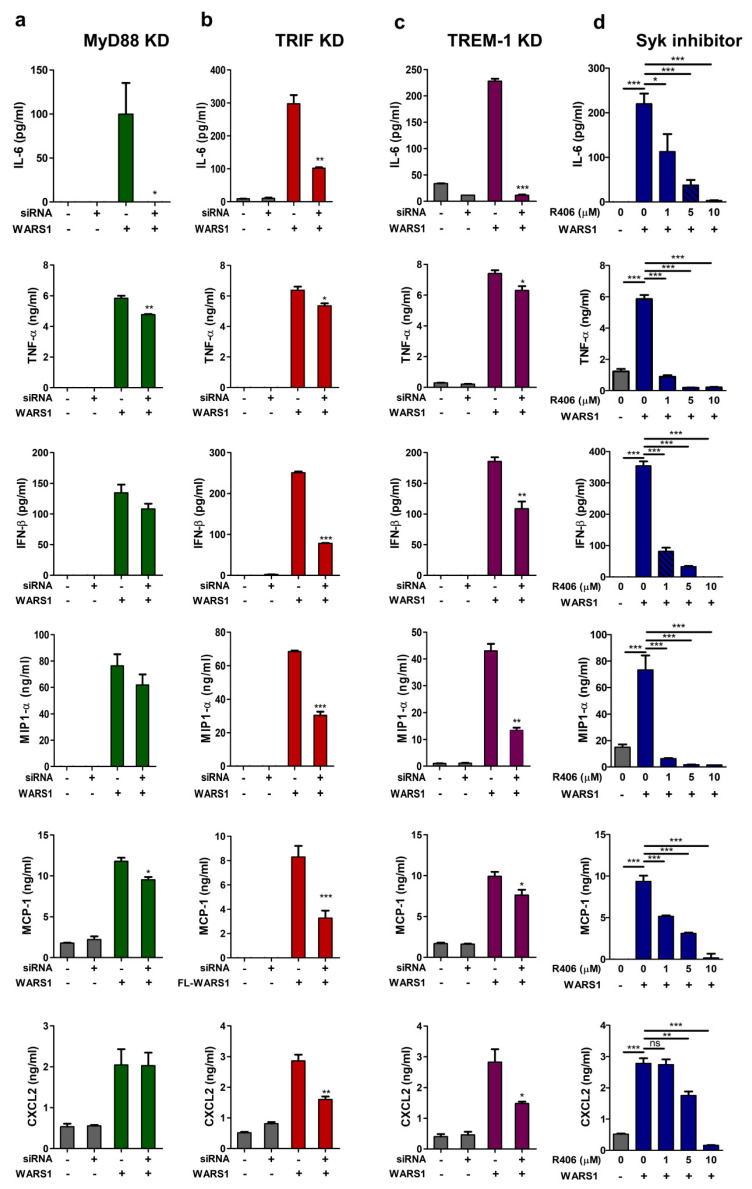
Cytokine/chemokine profiles mediated by MyD88, TRIF, TREM-1, and Syk in WARS1-treated macrophages. (**a**–**c**) J774.1 cells were transfected with 100 nM negative, (**a**) MyD88 siRNA, (**b**) TRIF siRNA, or (**c**) TREM-1 siRNA; 72 h after transfection, cells were treated with PBS or 100 nM WARS1 for 8 h, and then IL-6, TNF-α, IFN-β, MIP-1α, MCP-1, and CXCL2 levels in the supernatants were detected by ELISA. (**d**) J774.1 cells were treated simultaneously with increasing doses of the Syk inhibitor (R406) and PBS or 100 nM WARS1 for 8 h, and subsequently, MCP-1, TNF-α, IFN-β, IL-6, MIP-1α, MCP-1, and CXCL2 levels in the supernatants were detected by ELISA. (* *p* < 0.05, ** *p* < 0.01, *** *p* < 0.001).

**Figure 7 biomolecules-10-01283-f007:**
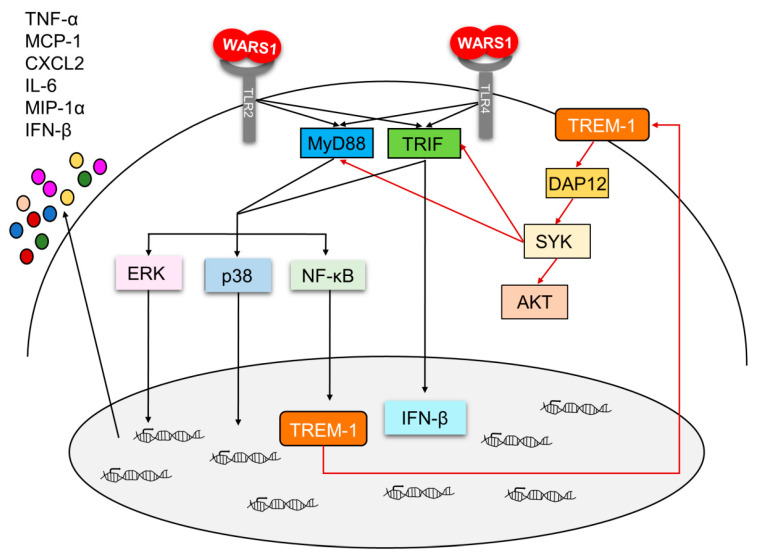
Summary of WARS1 signaling pathway in macrophages. WARS1 binding to TLR4 or TLR2 causes an activation of MyD88 and TRIF. MyD88 and TRIF increase TREM-1 expression and activate pro-inflammatory mediators that cause the release of cytokines and chemokines. TREM-1 receptor engages intracellular signaling molecules DAP12 and Syk, leading to a positive feedback on MyD88 and TRIF signaling.

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
