# Peer review of "Tryptophanyl-tRNA Synthetase 1 Signals Activate TREM-1 via TLR2 and TLR4"

_biomolecules, 2020, doi:10.3390/biom10091283_

Round 1

Reviewer 1 Report

The authors show that exogenous WARS1 (TrpRS) protein can increase the expression of TREM-1, a central regulator of inflammation. They show that the increased expression is dependent on TLR2/4 and Myd88 and/or TRIF signaling and activates TREM-1 downstream targets such as AKT and Syk. The work complements previous findings on the pro-inflammatory effect of WARS1. 

The authors do not specify whether experiments were performed repeatedly or only once. This should be clarified in the text. Some of the Western blots, especially in figure 5 are not convincing - if possible, they should be exchanged for clearer blots with less ambiguous exposure. The contrast in several blots should be readjusted.

2C: Why does only WARS induce the secretion of TREM1 and not LPS, despite LPS leading to higher cellular levels. What is the relevance of cellular vs secreted TREM1? The authors should include a discussion on this point and cite relevant literature.

2e: The roughness of the curves suggests that insufficient cell events were recorded. The authors should provide the gating strategy in the supplementary figures and state the number of cells.

2f: Figures are labeled as TREM1+DAPI even though it's brightfield or DIC and DAPI cannot be seen. Also, why does the cellular distribution differ so much between WARS1 and LPS treatment? Please explain/discuss.

3e: TLR2 knock down reduced TLR4 almost as much as the siRNA against TLR4 - why? Also, this observation is poorly reflected on protein level in the next figure.

4: Were Myd88 and TRIF on the original microarray? Were the same changes observed?

5c: The Western blots are not very clear - DAP-12 phosphorylation is hardly supported by this blot and the phospho-AKT blot looks strange: why is there a grey line between the lanes and the third lane so much darker? For several blots in figure 5 the exposure looks uneven. The following Western blots worry me: 5c p-DAP12, p-AKT, 5d MyD88, p-Syk, AKT, 5e: TRIF, p-Syk, p-Akt, p-IkB.

6: Is the cytokine signature of WARS1 expected to activate specific immune cells? The manuscript would benefit from a discussion of the biological details following exposure to WARS1 aside from the broad implication for sepsis.

In the discussion WARS1 is suggested as a potential sepsis treatment target but no obvious pathways towards this goal can be derived from this study - how do the authors propose to inhibit WARS1?

Endogenous WARS does not seem to play a role in this study. WARS is strongly upregulated upon cytokine exposure so it would be interesting to see what the effects of WARS knock down is in these cells. Indeed, it would be biologically interesting to see if WARS exposure increases endogenous WARS expression in a feed-forward loop.

Author Response

I appreciate the reviewer's valuable comments and suggestions. My responses are below. The revised text is indicated in yellow in the revised manuscript.

Reviewer 2 Report

This is an interesting study by Nguyen et al., in which the authors investigate the effect of WARS1 on TREM1 expression, signalling and amplification of TLR2/4 signalling cascade.

Comments to the authors:

  • please provide a description in the Methods of the cell stimulation with WARS1 and detail of the stimulation in the results description (duration and concentration) besides the description provided in figure legends;
  • are the concentration used 100nM WARS1 physiological? could the authors comment on this?
  • in paragraphs 3.2, 3.3 and 3.4, the authors investigate the effects of WARS1 in TREM1 expression (gene or protein) and not the intracellular singling activation, therefore it would be more correct to substitute the term "TREM1 activation" in the title of 3.2, 3.3 and 3.4 with "TREM1 expression"
  • In figure 4 the KD of myd88 is rather mild, did the author try to repeat this experiment with another siRNA? 
  • I do not completely agree with the conclusion of paragraph 3.6; from the  data of fig 6 I would conclude that WARS1-induced TREM1 pathway amplifies mainly TRIF pathways rather than MYD88. It would be important to repeat these assay with a double KD of TREM1&MYD88, TREM1&TRIF and the single KD to assess the relative contribution of single pathways and their synergetic effects
  • lastly fig3a shows no direct interaction of WARS1 with TREM1, so how does TREM1 get activated after being over expressed in response to TLR2/4 activation? did the authors check for secretion of HMGB1, PGLYRP1, Actin after WARS1 stimulation of macrophages?

Author Response

I appreciate the reviewer's valuable comments and suggestions. I have responded as detailed below. The revised text is highlighted in yellow in the revised manuscript.
